Does polyandry really pay off? The effects of multiple mating and number of fathers on morphological traits and survival in clutches of nesting green turtles at Tortuguero

Alfaro-Núñez Alonzo 1 alonzoalfaro@gmail.com
Jensen Michael P. 2
Abreu-Grobois F. Alberto 3
1 Center for GeoGenetics, Natural History Museum of Denmark, University of Copenhagen , Copenhagen , Denmark
2 Marine Mammal and Turtle Division, National Oceanic and Atmospheric Administration , La Jolla, CA , United States
3 Laboratorio de Genética, Unidad Académica Mazatlán, Instituto de Ciencias del Mar y Limnología, Universidad Nacional Autónoma de México , México
Ford Alex
Electronic publication date: 2015 Apr 7
Publication date: 2015
Volume: 3
Electronic Location ID: e880
Received 2014 Dec 19; Accepted 2015 Mar 17
Copyright: © 2015 Alfaro-Núñez et al.
Copyright year: 2015
Copyright holder: Alfaro-Núñez et al.
License: This is an open access article distributed under the terms of the Creative Commons Attribution License, which permits unrestricted use, distribution, reproduction and adaptation in any medium and for any purpose provided that it is properly attributed. For attribution, the original author(s), title, publication source (PeerJ) and either DOI or URL of the article must be cited.
License URL: https://creativecommons.org/licenses/by/4.0/

Keywords: Polyandry, Marine turtles, Mating systems, Evolution, Sperm competition, Paternal contribution, Population genetics, Microsatellites

Funding: University of Copenhagen This project study was mainly personally funded but with external support from the University of Copenhagen. The funders had no role in study design, data collection and analysis, decision to publish, or preparation of the manuscript.

==============================
Despite the long debate of whether or not multiple mating benefits the offspring, studies still show contradictory results. Multiple mating takes time and energy. Thus, if females fertilize their eggs with a single mating, why to mate more than once? We investigated and inferred paternal identity and number of sires in 12 clutches (240 hatchlings) of green turtles (Chelonia mydas) nests at Tortuguero, Costa Rica. Paternal alleles were inferred through comparison of maternal and hatchling genotypes, and indicated multiple paternity in at least 11 of the clutches (92%). The inferred average number of fathers was three (ranging from 1 to 5). Moreover, regression analyses were used to investigate for correlation of inferred clutch paternity with morphological traits of hatchlings fitness (emergence success, length, weight and crawling speed), the size of the mother, and an environmental variable (incubation temperature). We suggest and propose two different comparative approaches for evaluating morphological traits and clutch paternity, in order to infer greater offspring survival. First, clutches coded by the exact number of fathers and second by the exact paternal contribution (fathers who gives greater proportion of the offspring per nest). We found significant differences (P < 0.05) in clutches coded by the exact number of fathers for all morphological traits. A general tendency of higher values in offspring sired by two to three fathers was observed for the length and weight traits. However, emergence success and crawling speed showed different trends which unable us to reach any further conclusion. The second approach analysing the paternal contribution showed no significant difference (P > 0.05) for any of the traits. We conclude that multiple paternity does not provide any extra benefit in the morphological fitness traits or the survival of the offspring, when analysed following the proposed comparative statistical methods.

Introduction

Although the fertilization of eggs in many female animals is usually achieved through a single mating, multiple mating is nevertheless a relatively common observation in natural populations. While some authors have argued that its prevalence is intriguing due to the energetic fitness costs that such behaviour incurs (Lee & Hays, 2004; Bilde et al., 2009), others have demonstrated how females may gain indirect genetic benefits to offset the cost by mating with multiple males (Byrne & Roberts, 2000; LaBrecque et al., 2014). For example, polyandry may provide genetic benefits by improving the chance that females will acquire “good” genes or by enhancing the genetic diversity of their progeny (Yasui, 1997).

Traditionally, behavioural observations have been used to determine the mating patterns in many vertebrate and invertebrate species. However, the application of nuclear DNA markers as an alternate tool has transformed such research by providing direct insights into parentage in natural populations (Packer et al., 1991; Craighead et al., 1995; Keane, Dittus & Melnick, 1997; FitzSimmons, 1998; Uller & Olsson, 2008) and enabling investigation into the genetic consequences of multiple matings. For example, genetic polyandry of surveyed birds species occurs regularly in 86% of the species (Michl et al., 2002; Griffith, Owens & Thuman, 2002). As a result, sperm from different males may compete to fertilize a single clutch of eggs. This is the case in many species of insects, gastropods, fish, amphibian, birds and reptiles (FitzSimmons, 1998; Garcia-Vazquez, 2001; Jones & Clark, 2003; García-González & Simmons, 2005; Chandler & Zamudio, 2008; Beese et al., 2008; Noble, Keogh & Whiting, 2013; LaBrecque et al., 2014) or in progeny from the same brood in mammals (Clapham & Palsbøll, 1997; Shurtliff, Pearse & Rogers, 2005). Thus, sperm competition may be an important factor in the evolution of reproduction of many organisms, although the precise mechanism that determines sperm success is not fully understood (Jones & Clark, 2003).

As with all six other sea turtle species, female green turtles have been observed to undergo polygamous breeding (Pearse & Avise, 2001), that often result in multiple paternity (MP) among offspring from a single clutch. Both short term sperm storage and sperm competition have been proven to be important aspects of turtle mating system (FitzSimmons, 1998; Phillips et al., 2013), and as a result several hypotheses have been proposed regarding the benefits of this behaviour. These include increased offspring viability and genetic diversity, fertilization assurance and procurement of compatible gametes (FitzSimmons, 1998; Jennions & Petrie, 2000; Uller & Olsson, 2008). The adaptive value of polyandry as a mating strategy could be explained in terms of sperm competition, sperm storage and sperm degradation during the mating process, which occurs within female oviducts and/or the egg fertilization. As in many other reptiles species, sea turtles are capable of long-term sperm storage for several years (Ewing, 1943). However, sperm viability does decline drastically after time (Pearse & Avise, 2001). Most turtle species breed at intervals of two to five years (Galbraith, 1993), making any stored sperm from previous seasons highly valuable—due to its genetic variability across seasons—but most likely of low quality.

Previous studies have demonstrated that the frequency of MP varies both between (Moore & Ball, 2002; Hoekert et al., 2002; Crim et al., 2002; Theissinger et al., 2008; Phillips et al., 2013; Noble, Keogh & Whiting, 2013), and within species (Galbraith, 1993; FitzSimmons, 1998; Ireland et al., 2003; Jensen et al., 2006; LaBrecque et al., 2014). For green turtles, evidence of MP has been reported in several studies, but the proportion of clutches with MP varies among populations. For example, 9% of the clutches analysed in Australia (FitzSimmons, 1998), 50% of clutches in the Caribbean Tortuguero rookery (Peare & Parker, 1996), 61% of clutches on Ascension Island (Lee & Hays, 2004), and the highest documented frequency of MP at 75% in the Eastern Pacific (Pearse & Avise, 2001).

All this raises some important questions such as, why these large differences in MP occur among populations of the same species? Are there any measurable benefits to the offspring from MP such as increased hatching success, increased survival in the clutches and/or benefits in morphological traits (e.g., length, weight and crawling speed) when compared by the exact number of inferred fathers in the clutches? On the other hand, there may be other factors that might affect these morphological traits, and may confound the results (e.g., incubation temperature, rainfall or the size of the parents).

Whether MP results in fitness benefits to the species remains unclear. In a previous study from Ascension Island in the middle Atlantic, the fitness in clutches was found to be not correlated with the paternity (Lee & Hays, 2004). The measurements of fitness used were the size of the female, clutch size, proportion of eggs fertilized, proportion of eggs hatching and offspring survival. However, the authors of this study did not measure direct physical or morphological traits in the hatchlings. In the present study, most measures from the Lee & Hays (2004) study were included in the analysis. Moreover, in addition we expanded by including measures of morphological traits of offspring survival measured by the emergence success, weight, length and crawling speed in order to generate additional parameters that may provide a more detail insight into the offspring fitness. Furthermore, we incorporate and propose two different comparative approaches to evaluate clutches paternity effect in the offspring fitness, which are in more detail described below.

This study aims to determine the level of MP for the green turtle population nesting at Tortuguero, Costa Rica. Furthermore, we aim to provide a relative measure of fitness in terms of morphological traits of clutches and hatchlings into the wild, defined as emergence success, length, weight and crawling speed. This combination of measurements may (i) support previous results in which MP was found to provide NO benefits to offspring fitness, or (ii) contradict most studies by reaching new conclusions where MP may be found to provide direct morphological benefits at the initial stage of the sea turtles lifecycle.

Materials and Methods

Field sampling

Samples were collected in 2007 at Tortuguero (10°32′42.26″N, 83°30′11.35″W), on the Caribbean coast of Costa Rica. This is the largest green turtle rookery in the world that comprises 25% of the global abundance of nesting green females (Seminoff et al., 2014). Tortuguero is a 45 km long beach; this study was concentrated on the northern 2 km of the beach. Tissue samples were taken from the trailing edge of the left front flipper from 41 nesting females using a scalpel. Iodine was immediately applied to the wound to avoid infection. If not previously tagged, adult turtles were tagged with Inconel flipper tags in both front flippers to ensure identification. The minimum curved carapace length (CCL) was recorded using a measuring tape. The location of the nests on the beach was recorded by triangulation; information used 45 days later to track them down before hatching was expected. Additionally, smart iButton data loggers (ACR Systems Inc., Vancouver, Canada) were placed in small Ziploc plastic bags in the middle of the nest together with the eggs. Twelve smart iButtons were placed into different nests to register the temperature changes during the incubation process (5 data loggers were lost and never recovered). Temperature was recorded every hour in seven different nests during the entire incubation period (approximately 60 days). Information from the smart iButton data loggers was downloaded using the software acr smartbutton reader 1.32. To protect nests from fly infestation and restrain hatchlings after emergence, mesh cylinder cages covered with mosquito netting were placed over each nest. Nests were observed every two hours, seven days at week during the emergence phase to prevent hatchlings from overexposure to the sun. Tissue samples were taken from the posterior edge of the carapace from 530 hatchlings (from 12 different nests, averaging 45 hatchlings per nest). All tissue samples were preserved in a saturated NaCl with 20% DMSO (Dutton et al., 1999) at ambient temperature in the field, and thereafter shipped to our labs in Denmark and preserved at −20 °C in the laboratory.

Immediately after tissue collection, length, weight and crawling speed were recorded from each hatchling. Right after, the neonates were released. Offspring length and weight were measured using a 250 mm vernier calliper and a 50 g manual scale, respectively. Crawling speed was recorded by using a one-meter long tube (15 cm diameter) placed on the sand. Hatchlings were released on one side and allowed to crawl to the other end, and the time spent by hatchlings in this trajectory was used as a measure of the crawling speed.

Laboratory analysis

Microsatellites

A total of 297 turtles (41 adult females and 256 hatchlings) were used and analysed for variation at eight microsatellite loci in this study. Tissue samples were finely chopped and digested overnight with proteinase-K at 52 °C. Total genomic DNA was then extracted using Invisorb Spin Tissue Kit (Invitek Inc., Berlin, Germany) following the manufacturer’s instructions. The microsatellite primers (Table 1) used in this study were developed for different species of sea turtles; Cm3, Cm58, Cm72, Cm84, Cc117 (FitzSimmons, Moritz & Moore, 1995), Cc7 (FitzSimmons, 1998), Or-4 and Or-7 (Aggarwal et al., 2007). Forward primers were end-labelled with fluorescent dye TaqMan® (Applied Biosystems, Carlsbad, California, USA). Following the QIAGEN Multiplex PCR Handbook manufacturer instructions, the microsatellites were multiplexed in the amplification stage and onwards. Each PCR was carried out in a 10-µL reaction volume containing 1-µL of genomic DNA, 1-µL of primer mix (2 µM each primer), 3-µL RNase-free water and 5-µL of QIAGEN multiplex PCR master mix (provides a final concentration of 3 mM MgCl2). Amplification were carried out in a PXE 0.2 Thermal Cycler with an enzyme activation step at 95 °C for 15 min, followed by 30 cycles of denaturation at 94 °C for 30 °C s, annealing at 57 °C for 90 s and extension at 72 °C for 60 s, and a final extension step at 60 °C for 30 min. To check for contamination negative controls were included in all PCR runs.

Table 1 Eight different microsatellite loci, primer sequences where the forward primers were end-labelled with fluorescent dye TaqMan®, sea turtle species from which the primers were designed, annealing temperature, allele length, number of alleles (NA), expected heterozygosity (HE) and observed heterozygosity (HO) for 41 adult females sample size.

Locus	Primer sequence (5′ → 3′)	Species	Annealing temperature (°C)	Allele length (bp)	NA	HE	HO	
Cc117	TCTTTAACGTATCTCCTGTAGCTC	Caretta caretta	57	230–260	11	0.87	0.71	
	CAGTAGTGTCAGTTCATTGTTTCA							
Cc7	TGCATTGCTTGACCAATTAGTGAG	Caretta caretta	57	160–220	17	0.92	0.93	
	ACATGTATAGTTGAGGAGCAAGTG							
Cm3	AATACTACCATGAGATGGGATGTG	Chelonia mydas	57	154–198	10	0.75	0.63	
	ATTCTTTTCTCCATAAACAAGGCC							
Cm58	GCCTGCAGTACACTCGGTATTTAT	Chelonia mydas	57	124–156	8	0.63	0.61	
	TCAATGAAAGTGACAGGATGTACC							
Cm72	CTATAAGGAGAAAGCGTTAAGACA	Chelonia mydas	57	228–298	24	0.90	0.90	
	CCAAATTAGGATTACACAGCCAAC							
Cm84	TGTTTTGACATTAGTCCAGGATTG	Chelonia mydas	57	316–356	15	0.90	0.78	
	ATTGTTATAGCCTATTGTTCAGGA							
Or-4	AGGCACACTAACAGAGAACTTGG	Several species	52	81–125	13	0.88	0.88	
	GGGACCCTAAAATACCACAAGACA							
Or-7	GGGTTAGATATAGGAGGTGCTTGATGT	Several species	52	210–240	6	0.64	0.71	
	TCAGGATTAGCCAACAAGAGCAAAA							

After successful amplifications, 1-µL of each PCR product was mixed with LIZ 500 size standard and HI-DI Formamide mixture, and denatured at 94 °C for 4 min, snap-cooled on ice and loaded onto an ABI 3130 DNA sequencer (Applied Biosystems, Carlsbad, California, USA). PCR fragment lengths were scored using genemapper version 4.0 (Applied Biosystems, Carlsbad, California, USA). Using the DNA extracts of the 297 turtles, the procedure of PCR amplification and genotyping was repeated and no genotypic inconsistencies were found for any locus among replications (at least two replications were made for each sample).

Data analysis

Female population analysis

To assess the genetic diversity of the Tortuguero population the 41 female turtles were genotyped at eight microsatellite loci. micro-checker software version 2.2.3 (Van Oosterhout et al., 2004) was used to check the microsatellite data for null alleles, stutter errors, short allele dominance and allelic dropout. To estimate the allele frequencies and the frequency of null alleles identity4.0 software (Wagner & Sefc) was used. To determine if the data fit Hardy–Weinberg proportions, genepop version 3.3 (Raymond & Rousset, 1995) was used as implemented online at http://genepop.curtin.edu.au/genepop_op1.html. For this procedure, the Markov chain method with the default parameters suggested online was used (Guo & Thompson, 1992) (5,000 dememorizations with 1,000 batches and 5,000 iterations per batch). Fischer’s method for combining independent test results across loci was used. genepop was also used to test for genotypic linkage disequilibrium between loci using the standard Markov chain parameters of 1,000 dememorizations with 100 batches and 1,000 iterations per batch. Fischer’s method for combining independent test results across loci was used throughout.

Paternity analysis

Two microsatellite primers, Or-4 and Or-7, did not yield sufficiently reproducible PCR product in the hatchling’s DNA and were therefore excluded from the paternity analysis. Though, the mean number of hatchlings sampled per clutch at Tortuguero was 45, two clutches had a limited number of offspring (>25). Therefore, to prevent statistical artefact for the analyses, the sample size genotyped in this study was established to 20 offspring per clutch. Thus, twelve different nests and 20 random selected offspring from each nest were examined for paternity. Maternal and offspring genotypes were determined directly from the sampled females and hatchlings. To calculate the probability of detecting MP with one known parent, we used the prdm software (Neff & Pitcher, 2002). To determine the actual number offspring that are required to detect multiple sired broods with high probability (80 and 95%) the software takes into account: (i) number of loci; (ii) frequencies and number of alleles; and (iii) number of sires and reproductive skew. To test the power of detecting MP under different scenarios paternal contributions and number of fathers, different simulations using three different combinations of loci (Table 2) were carried out.

Table 2 Probability of detecting multiple paternity by using PrDM software (Neff & Pitcher, 2002).

Based on our baseline population frequencies, the model is used to determine the actual number of loci and offspring that are required to detect multiply mated broods with high probability (80 and 95%) and takes into account: (i) different number of loci; (ii) frequencies and number of alleles; and (iii) number of sires and reproductive skew. The three different combination of loci were, 8 loci, Cc117, Cc7, Cm3, Cm58, Cm72, Cm84, Or4 and Or7; 6 loci, Cc117, Cc7, Cm3, Cm58, Cm72 and Cm84; and finally 4 loci, Cc7, Cm3, Cm58 and Cm72.

Number of fathers	Combinations of loci	Paternal contribution	Number of offspring sampled	
			10	20	30	
2	8 loci	50:50	0.998	1.000	1.000	
6 loci	0.998	1.000	1.000	
4 loci	0.994	0.999	0.999	
8 loci	66:33	0.982	1.000	1.000	
6 loci	0.982	1.000	1.000	
4 loci	0.976	0.998	0.999	
8 loci	90:10	0.648	0.878	0.959	
6 loci	0.653	0.878	0.957	
4 loci	0.636	0.868	0.949	
3	8 loci	33:33:33	1.000	1.000	1.000	
6 loci	1.000	1.000	1.000	
4 loci	1.000	1.000	1.000	
8 loci	50:25:25	0.999	1.000	1.000	
6 loci	0.999	1.000	1.000	
4 loci	0.998	1.000	1.000	
8 loci	80:10:10	0.891	0.988	0.999	
6 loci	0.890	0.989	0.999	
4 loci	0.882	0.986	0.998	
4	8 loci	25:25:25:25	1.000	1.000	1.000	
6 loci	1.000	1.000	1.000	
4 loci	1.000	1.000	1.000	
8 loci	40:20:20:20	1.000	1.000	1.000	
6 loci	1.000	1.000	1.000	
4 loci	1.000	1.000	1.000	
8 loci	70:10:10:10	0.973	0.999	1.000	
6 loci	0.971	0.999	1.000	
4 loci	0.966	0.999	1.000	
5	8 loci	20:20:20:20:20	1.000	1.000	1.000	
6 loci	1.000	1.000	1.000	
4 loci	1.000	1.000	1.000	
8 loci	40:15:15:15:15	1.000	1.000	1.000	
6 loci	1.000	1.000	1.000	
4 loci	1.000	1.000	1.000	
8 loci	80:5:5:5:5	0.893	0.989	0.999	
6 loci	0.889	0.988	0.999	
4 loci	0.881	0.985	0.999	

To determine and to reconstruct the genotypes of the unknown fathers, we used software gerud2.0 (Jones, 2005). Paternal alleles were inferred from offspring genotypes once maternal alleles were determined. Alleles present in hatchlings that were different from the maternal alleles and, in addition, alleles that were homozygous in some hatchlings were considered to be paternal. In a diploid organism, any instance of more than two paternal alleles is an indication of multiple paternity. The inferred paternal alleles in a clutch were then tested in combination to determine which set of potential paternal genotypes could have produced the entire array. This approach produces multiple minimum-father solutions consistent with a given progeny array (parameters were setup to the maximum: 500,000 MaxNumSols; 2,000,000 MaxPPgens; 2,000,000 MaxGPgens and 200,000 MaxMaternalgens; AG Jones, pers. comm., 2012). gerud2.0 calculates relative likelihoods for each solution and picks the solution with the highest likelihood. The most likely minimum number of fathers for each clutch was calculated. The simulation package gerudsim2.0 (Jones, 2005) was used to test the reliability of gerud2.0 to correctly determine the number of sires and to correctly reconstruct their genotypes. Based on the allele frequencies of the markers being used the program simulated progeny arrays. This approach allows a simple assessment of confidence in the performance of gerud2.0.

Using the paternal genotypes inferred in gerud2.0, the probability that two fathers sharing a common genotype was calculated using the software genalex6.0 (Peakall & Smouse, 2006).

Statistical analysis

Once maternal, offspring and paternal genotypes for each nest had been inferred, paternal identity was assigned manually to each offspring. Thus, data was grouped by families containing mother and fatherhoods and morphological traits of hatchlings fitness (emergence success, length, weight and crawling speed).

The software r version 3.1.2 (R Development Core Team, 2008) was used to calculate the statistical regression analyses with random effects. The Pearson’s method was used for the correlation matrices. Results are presented for the additive mixed model as this model has one degree of freedom. ANOVA test and F-statistic were used to determine the significant effect between the inferred paternity and the morphological traits. The basic model used for the analysis was defined with the formula, y = α + β∗x1; where α is the intercept, and β is the fixed effect size for the covariate x1 (e.g., y = length, x = MP). The model also allows the use of multiple covariates, y = α + β1∗x1 + β2∗x2 + β3∗x3 + β4∗x4 + ⋯ βn∗xn (e.g., lme (length ∼ MP, random = ∼ 1| mother, data = data)); where length is notated as the dependent variable and MP as the independent variable. All models had the random family error effect for the mother, which must be considered due to the multiple nests. In this model there is always an assumption of causality, in the sense that we assumed that β will affect the variable y (e.g., MP should affect the offspring length). Any possible interactions between the variables were also checked. Two main comparative approaches were evaluated for the clutches paternity, where the one factor ANOVA and F-statistic were tested to determine the level of variation between the observations. The first approach evaluated clutches coded as the exact number of fathers, where nests were analysed assuming an additive effect by their number of inferred fathers on each nest (e.g. SP = 1 father; MP = 2 fathers, 3 fathers, 4 fathers and 5 fathers). Using the paternal genotypes inferred in gerud2.0, the second approach analysed the paternal contribution (fathers who gives greater proportion of the offspring), where offspring within each family is giving a value according to the paternal ranking of the proportion contributed (e.g. offspring of father who contributed the most within a nest is giving a value of 1, offspring of father who contributed the second highest within a nest is giving a value of 2, etc.).

Regression analyses were also performed by correlating the morphological traits (length, weight and crawling speed) with each other. Moreover, two other analyses were also made: (i) effect of the environmental variable (incubation temperature mean values) on morphological traits; and (ii) effect of mother size (CCL) on incubation temperature and on morphological traits.

Results

Female population analysis

The eight microsatellite loci varied in allele number (6–24) and in observed heterozygosity (0.61–0.90) (Table 1). The mean overall deviation from the Hardy–Weinberg proportions, FIS was 0.062. However, for two loci, Cc117 and Cm84 were relatively high (FIS = 0.200 and FIS = 0.144, respectively). All loci were in Hardy–Weinberg proportion (P > 0.05). Based on results from micro-checker2.2.3, the same two loci (Cc117 and Cm84) indicated homozygote excess, which suggests that null alleles may be present. In the remaining loci, there was no evidence for scoring error due to stuttering or allelic dropout. Estimated frequency of null alleles calculated in identity4. 0 showed also a relative high percentage for the same two loci (Cc117 : 8.7% and Cm84 : 6.2%). No genotypic linkage disequilibrium was detected between any loci (P > 0.05).

Paternity analysis

The paternity analysis was performed using the four microsatellite loci (Cc7, Cm3, Cm58 and Cm72) that amplified consistently and appeared most reliable (no evidence of null alleles). Assuming equal paternal contributions, the probability of detecting MP using prdm software with four loci was very high when sampling 20 offspring (PrDM = 0.999). The number of fathers and number of loci did not significantly affect this probability. A 66:33 skew or even 33:33:33 of paternal contribution had little apparent affect on PrDM (Table 2). However, a highly skewed paternal contribution of 80:10:10 and up to 90:10 did affect the PrDM relatively slight when 20 offspring were sampled. Furthermore, an estimate of paternal contribution showed that the average proportion of offspring sired by the “primary” male (male who gives greater proportion of the offspring) was higher than 50% across all MP clutches (see Fig. 1 labelled as father 1). In five nests, the primary father proportion of hatchlings was >80%, but only one was inferred to have just a single father (N07). The probability of detecting MP at the most skewed paternal contribution (90:10; two fathers) with four loci was PrDM = 0.868, so we assumed that the observation of SP in this nest was most likely correct.

Figure 1 Paternal contribution for all nests having multiple paternity (MP) and single paternity.

The different colours represent the proportion of offspring that each father has contributed per nest. The first colour in the bottom of each column represents the primary father or the father that contributed to most offspring until the last colour in the top representing the father with the small offspring contribution.

Thus, the null hypotheses of single paternity could not be rejected in only one nest (8%) and MP was found for the remaining 11 clutches (92%). The number of paternal alleles across all loci varied (from one up to seven alleles per locus) for the MP clutches. Between two to five fathers were inferred in the multiply sired clutches (Table 3) and each father contributed from one to 19 offspring (see Fig. 1). There was no evidence of fathers sharing the same genotype as no match was found across all loci nor at one locus. A total number of 35 different fathers were found to contribute to the paternity of the 12 different clutches. The allele frequencies obtained from identity 4.0 were used in the simulation package gerudsim2.0.

Table 3 Dataset of each nest analysed by mother length size (CCL), clutch size measured by the number of eggs, the number of hatchlings that emerged, the emergence success percentage, the incubation mean temperature registered in 7 different nests, the number of hatchlings genotyped.

The number of alleles and number of non-maternal alleles, paternal alleles (*) at the microsatellite loci Cc7, Cm3, Cm58 and Cm72. The evidence of multiple paternity and the minimum number of fathers inferred by the program GERUD 2.0.

Nest	Mother size CCL (cm)	Clutch size	Emerged hatchlings	Emergence success %	Incubation temperature (°C)	hatchlings genotyped	Cc7 (*)	Cm3 (*)	Cm58 (*)	Cm72 (*)	MP evidence	Minimum number of fathers	
N01	105.07	92	78	84.78%	–	20	3 (1)	3 (2)	2 (1)	4 (2)	Yes	3	
N02	104.13	97	91	93.81%	–	20	6 (4)	4 (3)	3 (2)	6 (4)	Yes	3	
N04	116.33	138	125	90.58%	30.91	20	4 (2)	2 (1)	3 (1)	4 (2)	Yes	2	
N05	105.20	128	114	89.06%	–	20	4 (2)	3 (2)	3 (2)	4 (2)	Yes	2	
N06	108.47	94	86	91.49%	31.77	20	4 (2)	3 (1)	3 (2)	6 (4)	Yes	3	
N07	107.37	96	82	85.42%	–	20	4 (2)	3 (1)	3 (1)	4 (2)	No	1	
N08	109.73	119	83	69.75%	–	20	3 (1)	3 (1)	4 (2)	4 (2)	Yes	2	
N10	106.63	115	106	92.17%	33.17	20	7 (5)	5 (3)	4 (2)	4 (2)	Yes	4	
N11	111.38	147	134	91.16%	29.73	20	5 (3)	3 (2)	4 (2)	9 (7)	Yes	5	
N12	114.77	104	92	88.46%	33.32	20	2 (1)	3 (2)	3 (2)	5 (3)	Yes	3	
N13	105.10	124	79	63.71%	32.01	20	3 (1)	3 (1)	3 (2)	4 (2)	Yes	2	
N14	110.47	97	86	88.66%	31.66	20	4 (3)	3 (1)	7 (5)	6 (4)	Yes	5	

Statistical analysis

There was a difference in the mean clutch size between multiply sired (114 ± 18.8) and singly sired (96 ± 0) clutch. The same was evident for the mean number of emerged hatchlings (MP = 99.6 ± 18.8 and SP = 82.0 ± 0). However, emergence success percentage (MP = 85.8% ± 9.5 and SP = 85.4% ± 0) showed to be almost identical.

Using the additive mixed regression model, the first comparative approach where clutches were grouped by the exact number of fathers, provided a significant difference (P < 0.05) for all morphological traits. The offspring’s weight and more so the length showed a tendency of comprising most of the higher values when sired by two and three fathers (see Fig. 2). However, the lowest values for the emergence success were recorded in nests sired by two fathers (Table 3). Box plot diagrams were made to graphically describe the distribution groups of the raw data through their five-father number summary (see Fig. 3). The diagrams reveal the tendency of higher values within the groups of two and three fathers for the length and weight traits; and so lower values in the groups of four and five fathers to be independently of the mother size. However, an exception to this pattern was observed for the crawling speed trait, which showed its highest values for the five fathers group (see Figs. 2 and 3). The number of observations (or hatchlings) per number of fathers was determined; 1 father = 20 (8.3%), 2 fathers = 80 (33.3%), 3 fathers = 80 (33.3%), 4 fathers = 20 (8.3%) and 5 fathers = 40 (16.6%). 66.6% of the observations were contained within the two and three fathers sub-groups.

Figure 2 Distributions of correlated morphological traits with 95% confidence intervals.

Distributions of length (A), weight (B) and speed (C) of green turtle hatchlings under categories specifying the exact number of inferred fathers on each nest (e.g., SP = 1 father; MP = 2 fathers, 3 fathers, 4 fathers and 5 fathers). The red lines indicate the 95% confidence interval. The diagrams revealed the tendency of higher values within the groups of two and three fathers for the length and weight traits. However, an exception to pattern was measured for the crawling speed (C) trait, which showed its highest values for the five fathers group. These results suggest a significant difference in fitness (as measured by our criteria) between hatchlings resulting from clutches fathered by one or more fathers.

The paternal contribution analysis where the association between proportion of clutch paternity and the morphological traits was assessed showed no significant difference (P > 0.05) for none of the parameters. Hence, primary fathers (1 single father) appeared not to confer fitness advantages to their specific offspring compared with secondary fathers (see Fig. 3).

Figure 3 Box plot of morphological traits correlations.

Box plot describing the raw data of length (A), weight (B) and crawling speed (C) by number of father groups observations through their five-number summaries, the smallest observation represented by the lowest line (sample minimum), lower quartile (Q1) 25% ≤ the lower line of the box, median (Q2) 50% of the observations ≤ the bold line into the box, upper quartile (Q3) 75% of the observations ≤ the upper line of the box, and largest observation (sample maximum) upper highest line.

The morphological traits of hatchlings fitness (length, weight and crawling speed) were plotted into linear regressions to determine the correlation between the three parameters per nest (see Fig. 4). Not surprisingly, there was a highly significant correlation between length and weight of (R2 = 0.97%; P < 0.001). However, non-significant correlation (P > 0.05) was found between the parameter crawling speed correlated with length and weight. These correlation analyses were in addition performed using all the 530 hatchlings (including all those that were not genotyped and hence not used to infer paternity), and the same pattern was found for this combined data set overall.

Figure 4 Linear regressions of morphological traits.

Linear regressions between offspring morphological traits in the next order: weight and length (A); crawling speed and length (B); and crawling speed and weight (C). Regressions were plotted for each nest, and P-values estimated in an overall. Regressions between weight (g) and length (mm) showed a high significant correlation (P < 0.001). The regressions between crawling speed and length (B) showed however in an overall non-significant correlation (P > 0.05). Furthermore, non-significant correlation (P > 0.05) was neither observed globally between crawling speed and weight (C) traits.

Incubation temperature was recorded for a random seven of the 12 clutches, all of which showed MP; consequently, it was not possible to test for differences against the unique SP nest. The highest temperature recorded was 37.0 °C in nest N12 during the last third time period of incubation. The lowest temperature was 26.0 °C recorded in both nests N04 and N11 in the beginning of the first third of incubation. The mean incubation temperature recorded for all nests was 31.8 °C (±1.2 °C).

The incubation temperature was found correlated with the emergence success, crawling speed and mother size (P < 0.005). Further, a highly significant correlation (R2 = 0.81; P < 0.001) was found between incubation temperature and mother size, while other morphological traits showed no correlation (P > 0.05). Mother size (as measured by the CCL) showed a highly significant correlation with the emergence success and crawling speed (P < 0.001). No significant correlation was found between mother size and any of the other morphological traits (P > 0.05).

Discussion

Multiple mating

The observation of a MP frequency of at least 92% in this study is the highest ever recorded for green turtles. There might be true differences in MP between green turtle populations, but methodological artefacts may also cause the discrepancy. For example, MP was previously reported for green turtles at Tortuguero with a frequency of 50% (Bjorndal, Bolten & Troëng). However, the previous study used only two microsatellites in two different nests with less than 15 offspring each and also used a different approach (UPGMA method). The power of inferring unknown parentage is based on number of markers, number of offspring and the allele frequencies of the population. Such power differences between studies only affect the probability of type II errors (the probability of false negatives), so we conclude that our very high frequency of MP is unlikely to be upwards biased. However, we are fully aware that the final use of only four microsatellite loci was low and it can potentially has its limitations and therefore we caution the general interpretation of the conclusion drawn in this study. From this and other studies, we can conclude that MP is not a rare or occasional event; it is in fact a general and widespread mating strategy for the green turtle with a variation in frequency amongst breeding population sizes. At present, no long-term study has been conducted to measure the frequency of MP within and across seasons of the same population, so it is hard to say whether this is a characteristic attribute for a given population.

This study examined a limited number of turtle clutches; 12 out of more than 170,000 estimated nests in Tortuguero in 2007 (Debade, del Aguila & Harrison, 2008). Moreover, the sampling of females was limited to the northern 2 km out of 45 km nesting beach. Green turtles have shown strong nesting site fidelity (Broderick et al., 2007) and females nesting in close proximity to each other at Tortuguero have been proven to be genetically closely related (R2 = 0.27%; P < 0.001) (Peare & Parker, 1996). However, our results from the genetic diversity analysis did not indicate a close relation between the females. It would be interesting to investigate further (within a coherent methodological framework) how the frequency of MP varies on a spatial and temporal scale incorporating further measurements of the mating process from also male turtles.

Most of our knowledge about female selecting between multiple males comes from experiments where two or more males are presented simultaneously to a female. Under natural conditions, however, females of many species rarely encounter potential mates simultaneously. This is due to mate choice often being constrained by time, mobility, predation pressure and multiple males fighting over a single female (commonly observed in green turtles). Thus the costs of comparing several potential mates can be considerably high (Klemme, Eccard & Ylönen, 2006; LaBrecque et al., 2014). We find it more likely that sea turtle female under breeding conditions might mate with the first male they encounter and they will probably mate again if another male appears and so on. Thus, we suggest multiple mating to be a common and general mating strategy in green turtles only limited by potential mate encounters and effective sex ratios at the breeding grounds.

The question here is which males sperm will fertilize most of the offspring. Several different theories have been proposed in this regard. Laboratory experiments in Drosophila have shown that later-mating males tend to father a greater proportion of the offspring and that there is a great variability among genotypes of males and of females in the magnitude of this later-male advantage (Clark, 1999; Stewart & Dutton, 2014). Similar analysis in spiders have also suggested that fertilization success should be biased towards later mates (Watson, 1991). FitzSimmons (1998) suggested that male green turtles that have successfully inseminated females with sufficient sperm might be out-competed by previous mates. If fertilization from previous season’s mating occurred, there is a possible loss of sperm through time in storage tubules, or older sperm may be less viable (Ewing, 1943; FitzSimmons, 1998). Equal male contribution by random sperm mixture could also occur. Nonetheless, unequal paternal contribution was observed in this analysis (see Fig. 1) and there appears to be a primary father siring ranging from 40% to more than 90% of the offspring in a clutch. This could be caused by sperm competition or simply by the fact that male turtle that had mated either first or last to fertilize the majority of the eggs but without providing any fitness improvement. Therefore, if there were no competition and just random mixing of sperm one would expect an even contribution of fathers.

Morphological fitness traits

Despite contradictory results from previous work (Jennions & Petrie, 2000; Fisher, Double & Moore, 2006; Jennions et al., 2007), the main conclusion in this study is that polyandry does not have a clear influence in the morphological fitness traits measured in green turtle offspring. The inferred average number of fathers was three (ranging from 1 to 5). This in returns suggests that most females successfully got fecundated by at least three males. Regardless of this tendency observed for clutches fathered by two to three different males which grouped the largest and heaviest offspring, we cannot ignore the fact that this may be entirely random as the same pattern was NOT followed by the other traits (e.g. emergence success and crawling speed). This is, in fact, the first time that effect of exact number of fathers has been assessed for a turtle population into the wild. In a manipulative mating experiment in small marsupials, offspring of polyandrous females (mated with exactly three males) were measured to grow faster than offspring of monogamous females (Fisher, Double & Moore, 2006), supporting the potential relevance of evaluating multiple mating on fitness by the exact number of fathers for other organisms as well. It has been suggested that even though multiple mating includes males that are of poorer quality and thus potentially decreasing the fitness of offspring, most of a female’s offspring would be sired by dominant high quality males (Klemme, Eccard & Ylönen, 2006). In theory, paternity should consistently be biased towards males with high fitness values. In other words, paternity should be biased towards males that elevate offspring performance (‘intrinsic male quality’ hypothesis). However, there could also be selection against males where paternal–maternal genome interactions will, in fact, lower the offspring performance (‘genetic compatibility’ hypothesis) (Jennions et al., 2007). These two hypotheses correspond to the contrast between additive (e.g., ‘good gene’) and nonadditive (e.g., dominance) genetics effects (Leal, 2001). To what extent do males vary predictably in their effect on offspring fitness? One male should satisfy the basic fertilization needs but without necessarily granting higher fitness. Thus, having multiple fathers might increase the chance of some fathers to produce stronger (fitter) offspring so that on average MP nests are more successful. In our study, the observed tendency of most of the nests to be fathered by two to three males and within those two father clusters offspring morphological traits (weight and length) were in average significant higher. This may suggest that there is a natural maximum size that may be explained in terms of sperm competition between males, which might ensure that a higher proportion of bigger offspring are produced. Nevertheless, this theory can be immediately overruled as the lowest emergence success values were recorded in nests sired by two fathers, bigger does not necessary provide fitter.

On the other hand, if sperm from four or more males interacted in the female oviducts, out-competition might occur resulting in a lower fitness (Pearse & Avise, 2001). We suggest that “best quality” sperm interaction may come out of the combination of two to three males to provide the “optimal” fitness in the offspring. Nonetheless, how precisely does this mechanist occur and work in sea turtles is still not known.

The paternal contribution analysis showed no correlations between none of the traits. Hence, we found no evidence of improved fitness in the offspring when sired by a primary father (male who gives greater proportion of the offspring) compared to any other secondary fathers. Similar results were concluded in Lee & Hays (2004) study.

In this study, measurements of morphological traits (crawling speed, length and weight) were taken following the same procedure. However, field and random error effects should be expected. Measurements between nests or families took place at different locations during different hours and even with different environmental conditions (e.g. rain, wet or dry sand, time of the day temperature, etc.) within those 2 km of the beach. Hence, we should expect variation and differences in offspring performance in terms of the measurement of fitness between nests defined for this study. For instances, it was observed a large variation in the crawling speed measures, the longer the offspring were retained, the slower they crawled probably because they become exhausted. The sand temperature and moisture, as well as the sampling time hour, seemed also to affect the offspring performance. All these may explain the great bias observed for the crawling speed, while length and weight showed not to be affected by retention time.

Environmental effect

The positive correlation found between mean incubation temperature and emergence success allows us to suspect that, within a certain range the higher incubation temperature increase the hatchling success as previous has already been proposed for green turtles and other reptile species (Lin et al., 2005; Burgess, Booth & Lanyon). Sexual determination in sea turtles is influenced by the temperature of the sand in which the eggs develop and sex is determined in the middle first third of the incubation (Wyneken, Godfrey & Bels, 2007; Stewart & Dutton, 2014). The incubation temperature that results in 50% of each sex is termed the pivotal temperature. For the green sea turtle the mean pivotal temperature is 28.8 °C (Mrosovsky, 1994). Nests with lower incubation temperatures will produce more males, whereas nests with higher temperatures will produce more females. Pivotal temperature is expected to differ between populations of the same species (Standora & Spotila, 1985; Mrosovsky, 1994; Stewart & Dutton, 2014; Godfrey & Mrosovsky). No study has reported the pivotal temperature for the Tortuguero population. Therefore, for this study 28.8 °C was considered the pivotal temperature for the green turtle species as this is the documented value for another relative close geographical site in Suriname (Godfrey & Mrosovsky; Kaska et al.). In this study, the mean temperature recorded in the middle first third period of the incubation was 30.2 °C (±1.2 °C). This suggests that there was a strong female-biased sex ratio of hatchling at the Tortuguero green turtle population during the 2007 season. The same result has been reported for other species of sea turtles in different nesting population, which has prompted concerns that global warming might be expected to skew the sex ratio towards females (Mrosovsky, 1994; Hays et al., 2003; Chaloupka, Kamezaki & Limpus, 2008). Under this scenario, if this leads to a very low proportion of males at the breeding grounds we could also suggest that MP, as a natural process, will be reduced in frequency causing a decline in hatchling’s emergence success. Recent evidence suggests that the interval between breeding seasons is less in male turtles than females (Hays, Mazaris & Schofield, 2014) and hence when female hatchlings dominate, operational (adult breeding) sex ratios are likely to me more balanced (Laloë et al., 2014). Hence MP may continue even when hatchling sex ratios are heavily female skewed.

The mean incubation temperature also showed a negative correlation with the size of the female turtle. This suggests that the larger females dig cooler nests. This can possibly be explained by the fact that larger females having longer hind flippers allowing the animal to dig deeper in the sand. Nevertheless, contradictory to our hypothesis that the deeper the eggs are laid the lower the mean temperature, it has been reported that nest depth has little influence on nest temperature (Van de Merwe, Ibrahim & Whittier, 2006; The Chu, Booth & Limpus, 2008).

In summary, all these lead us to conclude that in fact the incubation temperature factor has a great influence on the offspring morphological traits. If so, incubation temperature together with 2–3 fathers siring clutches may be important factors defining and perhaps shaping the morphological fitness traits of green turtle hatchlings in Tortuguero.

Supplemental Information

Supplemental Information 1 Female and offspring detected genotypes

Female and offspring detected genotypes for each of the four microsatellite loci -each nest is reported in an individual spreadsheet-. And determination of paternal contribution proportion per each individual nest.

Click here for additional data file.

Supplemental Information 2 Inferred parental genotypes

Inferred father genotypes per each individual nest using the final four microsatellite loci. And estimated amount of offspring that each father provided progenity to.

Click here for additional data file.

We would like to thank the following people and institutions for their important support and help during this project: Bethany Scott, Emma Harrison, Caribbean Conservation Corporation and their personnel, Federico Bolaños, Gerardo Chaves, Volker Loeschcke and Hans R. Siegismund. To Eva-Maria Didden, Ximena Velez-Suazo and Anders Albrechtsen for their technical support with the statistical data analysis. Special thanks to M. Thomas P. Gilbert, Kyle Van Houtan, Rasmus Heller, Greame Hays and one anonymous reviewer for the comments and improvements made to an earlier version of this manuscript.

Additional Information and Declarations

Competing Interests

Author Contributions

Animal Ethics

The authors declare there are no competing interests. However, we clarify that Michael P. Jensen is an employee of the National Oceanic and Atmospheric Administration, which is a non-academic affiliation.

Alonzo Alfaro-Núñez conceived and designed the experiments, performed the experiments, analyzed the data, contributed reagents/materials/analysis tools, wrote the paper, prepared figures and/or tables, reviewed drafts of the paper, collected samples.

Michael P. Jensen and F. Alberto Abreu-Grobois conceived and designed the experiments, wrote the paper, reviewed drafts of the paper.

The following information was supplied relating to ethical approvals (i.e., approving body and any reference numbers):

All tissue skin samples were collected in strict accordance with the recommendations in the Guide for the Care and Use of Laboratory Animals of the National Institutes of Health (Eight Edition, 2011) and exported under relevant CITES permits (host institute permit DK03). Moreover, samples were originated from live animals following a protocol made specially and approved by Universidad de Costa Rica (UCR) in cooperation with the Costa Rican Ministry of Environment and Energy (MINAE).

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
