# Peer review of "Does polyandry really pay off? The effects of multiple mating and number of fathers on morphological traits and survival in clutches of nesting green turtles at Tortuguero"

_PeerJ, doi:10.7717/peerj.880_

## Round 0.1 · original submission · Minor Revisions

Dear authors,

Thank you for submitting to PeerJ. Please find enclosed the comments from the two reviewers.

·

Basic reporting

Here tissue samples are collected from nesting female green turtles and their hatchlings at an important nesting beach in Costa Rica. The incidence of multiple paternity is then assessed within clutches and this incidence is compared to traits of the hatchlings such as their size and crawling speed. It is concluded that the incubation environment may be more important that the incidence of multiple paternity in influencing hatchling fitness.

This is an interesting manuscript that I enjoyed reading. There has been heated debate concerning whether multiple paternity confers fitness benefits to offspring. This manuscripts provides another important piece of this puzzle. Given the broad interest in this topic this manuscript is likely to create interest. Throughout the writing and presentation are thorough. I only have some minor comments that will be straightforward for the authors to address.

1. Line 120. “pack” not “back”

2 Line 446. In this section of implications of climate change I would add one or two lines to say that recent evidence suggests that the interval between breeding seasons is less in male turtles than females (Frontiers Mar. Sci. 2014) and hence when female hatchlings dominate, operational (adult breeding) sex ratios are likely to me more balanced (Nature Climate Change 2014). Hence MP may continue even when hatchling sex ratios are heavily female skewed. This will bring this section of the Discussion up to date with the current state of the art in this area. See :

Hays GC et al. Different male vs. female breeding periodicity helps mitigate offspring sex ratio skews in sea turtles. Front. Mar. Sci. 1:43. doi:10.3389/fmars.2014.00043

Laloë J-O et al. (2014). Effects of rising temperature on the viability of an important sea turtle rookery. Nature Climate Change 4, 513-518. doi: 10.1038/NCLIMATE2236

4. Line 449 “allowing”

5. Fig. 4. I would only include a line in a plot where there is a significant relationship.

In summary a nice manuscript that moves this area forward with some interesting data.
Graeme Hays, 27/12/2014

Experimental design

See above

Validity of the findings

See above

Additional comments

See above

Reviewer 2 ·

Basic reporting

This study provides improved data for the level of the multiple paternity (MP) of the green turtle population in Costa Rica. Genotypes of the fathers for 12 clutches were determined, and on average three fathers per clutch were estimated. In addition, authors investigated the relationships between MP and the hatching success, morphological traits of hatchlings, and incubation temperature. They concluded MP does not provide any benefit to the morphological traits of hatchlings. The approach is interesting, and their findings are valuable. However, I would have comments and suggestions.

Authors expand the previous study by Lee & Hays (2004) by the additional measures of physical and morphological traits of hatchlings. However, in the Introduction it is not clear what will be clarified by those additional components or why they need to the additional measures. Also, it seems that authors originally assume that the morphological traits measured in this study can be the indicators of fitness in some descriptions. However, they discuss the availability of the morphological traits as indicator of fitness. Therefore, overall, it is confusing for me that this study (or each result) focuses on either the influence of MP (the number of fathers per clutch) to the morphological traits of hatchlings or the evaluation of MP (the number of fathers per clutch) to the fitness of hatchlings. Although several questions are arisen in the second last paragraph in the Introduction, the questions should be simple and specific according to what is investigated in this study.
I think that it needs to restructure Lines 89-107 of the Introduction to present more explicit objectives (or their hypothesis) of this study.
I would also suggest to re-edit the Discussion, especially the “Fitness” chapter of the Discussion. Discussion along with the ovjectives presented in the Introduction would make the paper more understandable and interesting.

Experimental design

I am not familiar with the software used for the paternity analysis and the statistical tests in this study, so cannot comment on their appropriateness. However, I have some comments.

The relationships between the mother size and the length and weight of the hatchlings should be firstly analyzed. As a result there was no significant relationship between them, but it seems more natural to evaluate the effect of MP to the size of hatchlings after the possibility that the mother size influence the size of offspring is denied.

I didn’t understand how the genotyped 20 individuals were chosen from the total hatchlings of each clutch.

The information that the “mean values” of the incubation temperature were used for regression analyses should be described in the Materials & Methods section.

Validity of the findings

No Comments

Additional comments

Specific comments:
L91-95: “Moreover, in addition…” This sentence lacks readability.

Following sentences in results section should be transferred to the Materials & Methods.
L271-272: “The allele frequencies obtained from IDENTITY4.0 were…”
L282-284: “Box plot diagrams were…”
L296-298: “The morphological traits of hatchlings fitness (length, weight and crawling speed) were plotted…”

L242: Please add “(table 1)” after the sentence “The eight microsatellite loci varied in allele number…”
L281-282 Please add “(table 3)” after the sentence “However, the lowest values for the emergence success were recorded in nests sired by two fathers”.

L371: “Despite contradictory results from previous work” I do not understand which work is supposed here. Please describe in detail.

Figure 1 The legend of the figure 1 noted as “farther 1”, “father2”… should be revised to as “primary”, “second(ary)”…to fit the decription of the manuscript.

---

## Round 0.2 · accepted · Accept

Dear Dr Alfaro-Núñez,

Thank you once again for submitting your manuscript to PeerJ. Please note that when the proofs come through I recommend changing the word "contradicting" to "contradictory" in the 1st sentence of the abstract.